# High Molecular Weight Fucoidan Restores Intestinal Integrity by Regulating Inflammation and Tight Junction Loss Induced by Methylglyoxal-Derived Hydroimidazolone-1

**DOI:** 10.3390/md20090580

**Published:** 2022-09-17

**Authors:** Jae-Min Lim, Hee Joon Yoo, Kwang-Won Lee

**Affiliations:** 1Department of Biotechnology, College of Life Science and Biotechnology, Korea University, Seoul 02841, Korea; 2Department of Food Bioscience and Technology, College of Life Science and Biotechnology, Korea University, Seoul 02841, Korea

**Keywords:** fucoidan, intestinal barrier, advanced glycation end product, methylglyoxal-derived hydroimidazolone-1

## Abstract

Fucoidan from brown seaweeds has several biological effects, including preserving intestinal integrity. To investigate the intestinal protective properties of high molecular weight fucoidan (HMWF) from *Undaria pinnatifida* on intestinal integrity dysfunction caused by methylglyoxal-derived hydroimidazolone-1 (MG-H1), one of the dietary advanced-glycation end products (dAGEs) in the human-colon carcinoma-cell line (Caco-2) cells and ICR mice. According to research, dAGEs may damage the intestinal barrier by increasing gut permeability. The findings of the study showed that HMWF + MG-H1 treatment reduced by 16.8% the amount of reactive oxygen species generated by MG-H1 treatment alone. Furthermore, HMWF + MGH-1 treatment reduced MG-H1-induced monolayer integrity disruption, as measured by alterations in transepithelial electrical resistance (135% vs. 75.5%) and fluorescein isothiocyanate incorporation (1.40 × 10^−6^ cm/s vs. 3.80 cm/s). HMWF treatment prevented the MG-H1-induced expression of tight junction markers, including zonula occludens-1, occludin, and claudin-1 in Caco-2 cells and mouse colon tissues at the mRNA and protein level. Also, in Caco-2 and MG-H1-treated mice, HMWF plays an important role in preventing receptor for AGEs (RAGE)-mediated intestinal damage. In addition, HMWF inhibited the nuclear factor kappa B activation and its target genes leading to intestinal inflammation. These findings suggest that HMWF with price competitiveness could play an important role in preventing AGEs-induced intestinal barrier dysfunction.

## 1. Introduction

Intestinal barrier dysfunction, often called “leaky gut”, makes individuals vulnerable to various chronic inflammatory intestinal disorders, including diabetes and inflammatory bowel disease (IBD) [1]. Intestinal barrier function exerts a pivotal role for selective intestinal permeability that blocks the influx of food-derived toxicants and bacteria or their toxic components such as lipopolysaccharide (LPS) into the bloodstream, which can evoke inflammatory responses and tissue damage. Intestinal barrier integrity is chiefly maintained by a monolayer of intestinal epithelium sealed together by intracellular junction complexes such as tight junction (TJ) and adherens junction [2]. Since the zonula occludens-1 (ZO-1), occludin, and claudin complexes in the intestinal epithelium are important for paracellular permeability, it is crucial for TJ proteins to interact and form tight connections in order to maintain an intact intestinal barrier [3]. Thus, changes in TJ proteins may result in modifications to intestinal integration.

Advanced glycation end products (AGEs) can be formed by two pathways: endogenous and exogenous. Diet is the main source of exogenous AGEs due to the Maillard reaction in food components and processing conditions such as high temperature [4]. It is reported that dietary AGEs (dAGEs) ingestion contributes to AGEs measured in plasma and organs in humans and animals [5,6]. Ingested dAGEs can trigger negative health effects including inflammation, endothelial dysfunction, and diabetic complications [7]. Previous study has shown that consuming a heat-treated diet containing dAGEs for 24 weeks increases gut permeability, suggesting that dAGEs damage the intestinal barrier [8]. It has been reported that the Western diet, which has a relatively high proportion of sugar and fat, is linked to an increase in IBD incidences, hence increased intake of dAGEs is indicated [9]. Furthermore, dAGEs contribute to circulating AGEs and the AGEs pool in vivo, indicating that dAGEs can influence endogenous AGEs formation through AGEs metabolism [10]. Among well-characterized AGEs, methylglyoxal-derived hydroimidazolone-1 (MG-H1) was detected as having higher levels in heat-processed foods than *N*^ε^-(Carboxymethyl)lysine (CML) and *N*^ε^-(Carboxyethyl)lysine (CEL) [11]. When AGEs bind to the receptor for AGEs (RAGE), the AGE/RAGE axis can induce various diseases by activating intracellular pathways that promote oxidative stress through generating reactive oxygen species (ROS) and inflammation-causing cellular damage [12]. In particular, the structural characteristic of MG-H1 of binding tightly to RAGE can contribute to IBD [13,14]. However, the involvement of MG-H1/RAGE axis in triggering ROS production, leading to intestinal diseases, has not been fully studied.

Fucoidan is a bioactive polysaccharide extracted from brown algae and composed of sulfate, uronic acid, and sugars including galactose, and fucose [15]. Fucoidan has various biological properties associated with immune homeostasis, anti-inflammation, anti-oxidation and intestinal protection [16]. It is reported that bioactivities of fucoidan are sensitive to changes in their structural compositions, molecular weight (MW), species, extraction methods and fractionation methods [17,18]. *Undaria pinnatifida* (*U. pinnatifida*), which is widely distributed in the sea of Asian countries, is commonly consumed in Korean diets [19]. Recently, studies on regulating gut health of fucoidan derived from the sporophyll of *U. pinnatifida* have been reported [20,21,22]. In particular, the immunomodulating activity of high molecular weight fucoidan (HMWF) from *U. pinnatifida* has been reported [23]. However, the intestinal protective activities against MG-H1-induced intestinal barrier disruption have not been studied. As a result, we investigated the intestinal restorative potential of HMWF on intestinal damage generated by the MG-H1/RAGE axis in intestinal epithelial cells and an animal model.

## 2. Results

### 2.1. HMWF Treatment Prevents the Cytotoxic Effects of MG-H1 in Caco-2 Cells

To investigate the cytotoxicity of HMWF in the human colon carcinoma cell line (Caco-2) cells, 3-[4,5-dimethylthiazol-2-yl]-2,5-diphenyltetrazolium bromide (MTT) assay was performed with various concentrations of HMWF. Caco-2 cells were treated with HMWF (0–1000 μg/mL) for 24 h. There was a significant (*p* < 0.05) reduction in cell viability at 500 μg/mL (90%) and 1000 μg/mL (80%) compared with the control group (100%) (Appendix A). As cell viability percentages above 80% are regarded as non-toxic [24], and the highest absorption rate of fucoidan in Caco-2 cells was demonstrated at a concentration of 500 μg/mL [25]. Taken together, 500 μg/mL HMWF was employed in subsequent experiments.

When Caco-2 cells were treated with MG-H1 and HMWF concurrently, we evaluated cell viability and ROS generation to further determine how HMWF protects the cells from the cellular stress caused by MG-H1. There was no significant reduction in cell viability in the control, MG-H1, or MG-H1 + HMWF groups, as shown in Figure 1A. Since ROS triggered by dAGEs can affect intestinal inflammation, 2,7-dichlorofluorescin diacetate.

(DCFH-DA) assay was also conducted to assess whether HMWF treatment inhibit ROS production. Figure 1B showed that the incubation with MG-H1 significantly (*p <* 0.05) (354.6 ± 10.9%) increased intracellular ROS in comparison to the control, whereas HMWF treatment inhibited ROS production significantly (294.9 ± 3.2%) (*p* < 0.05) compared to MG-H1 treatment alone. Thus, HMWF treatment can contribute to the intestinal stability by inhibiting oxidative stress.

### 2.2. HMWF Treatment Restores Intestinal Barrier Fucntion against MG-H1-Induced Intestinal Damage in Caco-2 Monolayers

To evaluate the protective effects of HMWF against intestinal barrier dysfunction caused using MG-H1 treatment in differentiated Caco-2 cell monolayers, the transepithelial electrical resistance (TEER) was measured (Figure 2A). Initial TEER values were measured before 500 μg/mL HMWF and 10 μM MG-H1 were treated to the apical side of the Caco-2 cell monolayer. The TEER values were once more measured 24 h after the treatment. The cells treated with MG-H1 showed a significant (*p* < 0.01) decrease in the integrity of Caco-2 monolayers compared with the untreated control cells as TEER values were diminished to (75.5 ± 9.1%) of that of the control. However, the TEER value after HMWF + MG-H1 treatment increased significantly (*p* < 0.001) to (135.7 ± 4.9%) as compared to MG-H1 treatment, protecting the monolayer integrity in Caco-2 cells (Figure 2A).

Additionally, with the control, MG-H1, and MG-H1 + HMWF treatment, the permeability coefficient (Papp) was (1.32 ± 0.03, 3.80 ± 0.10, and 1.40 ± 0.04) × 10^−6^ cm/s, respectively (Figure 2B). Thus, paracellular permeability was significantly (*p* < 0.001) increased in MG-H1 treated cells compared with the control cells and significantly (*p* < 0.001) decreased in MG-H1 + HMWF treated cells when compared to MG-H1 treated cells.

### 2.3. HMWF Administeration Attenuates Colon Damage in Mice Given MG-H1 Intravenously

HWMF was given to the low dose (L-HMWF) and high dose (H-HMWF) in this investigation at doses of 25 and 75 mg/kg body weight (b.w.), respectively. In the previous experiment, both doses of HMWF treatments had immunostimulant effects [23]. The current results showed that the colon lengths among all groups showed no significant difference (Figure 3A,B). After oral administration of FITC-dextran, the serum concentration of FITC-dextran in the control group was 10.5 μg/mL; however, there was no significant difference in FITC-dextran levels between the MG-H1 treated group and the MG-H1 + HMWF treated groups (Figure 3C). To evaluate the histopathological changes of colon tissues, we performed hematoxylin and eosin staining. No epithelial damage or inflammatory cell infiltration was seen in the control group (Figure 3D). The MG-H1 group may suggest the lymphoid aggregate in colon tissue which means infiltration of neutrophils (black arrow). The lymphoid aggregates might be effectively diminished in the MG-H1 + HMWF treated groups. It should be noted, however, that H&E staining is not conclusive for neutrophils, and no staining for a neutrophil specific marker such as RB6-8C5 was conducted in this study. The current data are qualitative in nature. MG-H1 injection increased myeloperoxidase (MPO) activity (28.1 ± 16.5 U/g tissue). Although there was no statistically significant difference from the MG-H1 treated group, HMWF treatment decreased the MG-H1-induced MPO activity (19.8 ± 18.2 U/g tissue in L-HMWF group) and 23.4 ± 8.9 U/g tissue in H-HMWF group) (Figure 3E). Thus, HMWF treatment may inhibit the MPO-mediated oxidative stress and colon inflammation induced by MG-H1.

### 2.4. HMWF Treatment Inhibits TJ Loss Caused by MG-H1 in Caco-2 Cells and Mice

It is thought that intestinal integrity is connected to the expression of TJ proteins. To demonstrate the protective effects of HMWF on TJ damage induced by MG-H1 treatment, the mRNA and protein expression levels of TJ markers such as ZO-1, occludin, and claudin-1 were determined. Cells were treated with MG-H1 and HMWF under the same conditions as with the previous experiments. MG-H1 treatment significantly (at least *p* < 0.05) reduced the expression of TJ markers at the mRNA and protein levels when compared to the control (Figure 4A,C). However, MG-H1 + HMWF treatment significantly (0.46 ± 0.00) (*p* < 0.001) increased occludin mRNA expression and significantly (0.84 ± 0.09) (*p* < 0.05) elevated claudin-1 protein expression. These observations indicate HMWF treatment may restore expression of TJ proteins and be associated with intestinal integrity enhancement.

TJ marker expressions at the mRNA and protein levels in mouse colon tissues determined whether HMWF administration inhibits MG-H1-induced TJ disruption. The mRNA expressions of TJ markers were significantly decreased (0.63 ± 0.14 and 0.69 ± 0.27 in occludin and claudin-1 respectively) (*p* < 0.05) using intravenous infusion of MG-H1 compared to the control (Figure 4B). The mRNA expression of ZO-1 was significantly restored (0.88 ± 0.17) (*p* < 0.05) with H-HMWF administration (75 mg/kg b.w.). As shown in Figure 4D, in accordance with this tendency, treatment with MG-H1 significantly reduced (*p* < 0.001) the TJ markers’ ZO-1 and occludin protein expression. Despite there being no significant difference, mice treated with HMWF treatment had higher expression of the proteins ZO-1 and occludin in their colons. In addition, the location of occludin was investigated to see the disruption of the TJ structure in Caco-2 cells (Figure 5). In Caco-2 control cells, occludin was localized to TJ in a “honeycomb” pattern. Whereas the treatment of MG-H1 weakened clear continuity, the combined MG-H1 + HMWF treatment resulted in a pattern similar to control cells. These results suggest that HMWF treatment might restore intestinal barrier dysfunction by restoring TJ loss caused by MG-H1 treatment.

### 2.5. HMWF Treatment Lowers RAGE and Inflammatory Mediators in MG-H1-Treated Caco-2 Cells and Mice

Because binding AGEs to RAGE can activate cellular signaling pathways, resulting in intestinal inflammation and illness [12], it was investigated if MG-H1 treatment increases RAGE mRNA and protein expression in Caco-2 cells and mice. RAGE expression was significantly (1.49 ± 0.08 and 1.60 ± 0.35, respectively) (at least *p* < 0.05) increased at both the mRNA and protein levels in MG-H1-exposed Caco-2 cells (Figure 6A–C). HMWF treatment, on the other hand, significantly (0.94 ± 0.24 and 0.74 ± 0.33, respectively) (*p* < 0.05) prevented the upregulation of RAGE expression at the mRNA and protein levels caused using MG-H1 treatment. Similar results were seen in an in vivo experiment, in which administration of HMWF significantly (1.74 ± 0.65 and 1.80 ± 0.47, respectively) (*p* < 0.05) reduced the increased expression of RAGE protein induced using MG-H1 treatment in mouse colon tissues Figure 6D in both concentrations. Our results imply that HMWF treatment could play a significant role in mediating MG-H1-induced RAGE expression.

To determine the impact of HMWF on the inflammation induced by MG-H1 in Caco-2 cells and mouse colon, the expression of pro-inflammatory cytokines was measured at the mRNA level. Interleukin 6 (IL-6) and tumor necrosis factor (TNF-α) expression was higher in the MG-H1 treated cells than in the control cells, with the former being significantly increased (1.65 ± 0.26 and 1.21 ± 0.21, respectively) (*p* < 0.05) (Figure 6A). However, the combined MG-H1 + HMWF treated cells showed lower levels of those cytokines than the MG-H1 treated cells, and even less than the control cells (0.49 ± 0.04 and 0.61 ± 0.12, respectively) (*p* < 0.05). In the mouse colon tissues, TNF-α expression is increased in the MG-H1 treated group compared to the control group (1.18 ± 0.22) (Figure 6B). HMWF oral treatment, on the other hand, significantly reversed (0.82 ± 0.36 in low concentration) (*p* < 0.05) the TNF-α elevation caused by MG-H1 intravenous infusion. As a result, HMWF treatment might well be able to suppress the increase in pro-inflammatory cytokine expression seen in inflammatory bowel disease.

### 2.6. HMWF Treatment Inhibits RAGE-Mediated NF-κB Signaling in Intestinal Inflammation

We investigated the impact of HMWF and FPS-ZM1 (a selective RAGE inhibitor) on the MG-H1-induced TJ disruption and phosphorylation of nuclear factor kappa B (NF-κB) in order to better understand the molecular mechanisms underlying RAGE-induced intestinal inflammation. The pre-treatment with FPS-ZM1 as well as the HMWF treatment, as seen in Figure 7A, restored the protein expression of TJ indicators such as ZO-1 and occludin. Because FPS-ZM1 pre-treatment (1.16 ± 0.19) as well as HMWF treatment significantly reduced (1.09 ± 0.12) (*p* < 0.05) RAGE protein expression and the ratio of phospho-NF-κB expression (p-p65) to that of NF-κB p65 (0.92 ± 0.04), which was significantly increased (*p* < 0.05) by MG-H1 treatment (1.21 ± 0.04) (Figure 7B).

Additionally, it was demonstrated that pro-inflammatory cytokines such cyclooxygenase-2 (COX-2) and inducible nitric oxide synthase (iNOS) were expressed in higher amounts in Caco-2 cells after being treated with MG-H1 (2.11 ± 0.30 and 1.78 ± 0.07, respectively) (*p* < 0.01) (Figure 7B). However, HMWF treatment as well as FPS-ZM1 pre-treatment reduced COX-2 and iNOS protein expression (1.15 ± 0.24 and 1.01 ± 0.20, respectively) (*p* < 0.05). These findings suggest that HMWF inhibited NF-κB activation resulting in inflammation by preventing MG-H1 from binding to RAGE in the same way that FPS-ZM1 did. Therefore, by binding to RAGE, MG-H1 may activate NF-κB, which in turn causes an up-regulation of pro-inflammatory mediators. In addition, the distribution of occludin protein in Caco-2 monolayers after the FPS-ZM1 pre-treatment as well as HMWF treatment, is restored (Figure 7C). This shows that RAGE may be involved in MG-HI-induced TJ dysfunction.

## 3. Discussion

The Maillard reaction facilitates the synthesis of dAGEs in meals containing proteins and carbohydrates during food processing, such as sterilizing and baking, and the possible detrimental health consequences of dAGEs on human health have been studied [7]. Previous studies showed that dAGEs have systemic effects on human health [26,27,28,29]. After ingesting dAGEs, they reach systemic circulation via the human gastrointestinal tract, hence the intestinal barrier can act as a checkpoint for dAGE-induced systemic effects. Among the AGEs, MG-H1 has a high level in processed foods, with 15 to 60 mg in 100 g of processed foods [11]. According to previous studies, 10 to 30% of orally ingested dAGEs is absorbed into systemic circulation [30,31]. As a result, 5 to 20 mg of MG-H1 might be absorbed when 100 g of processed foods are ingested. The volume of circulating blood within an individual varies with b.w. and size but the average volume of circulating blood is estimated to 5 L [32]. Given the molecular weight of MG-H1, we estimated that MG-H1 levels in circulating blood varied between 3.5 μM and 14.1 μM. In addition to this, blood MG-H1 levels in people who eat processed foods and have insulin resistance ranges from 2.69 μM to 11.33 μM [33]. In this investigation, the experiment was carried out with MG-H1 at a concentration of 10 μM.

Fucoidan is a potential bioactive substance extracted from brown algae used for disease prevention and treatment, and the critical bioactive ingredient comprises the sulfated group [34]. In the case of the MTT experiment, the cell viability decreased in a concentration-dependent manner from the concentration of 50 μg/mL for fucoidan from *S. cinereum* [35], but in the case of HMWF, the cell viability was maintained at more than 90% up to the concentration of 500 μg/mL. Fucoidan isolated from the brown algae *Sargassum cinereum* had a sulfate group concentration of 3.7% [35], whereas HMWF produced from *Undaria pinnatifida* had a sulfate group value of 30.9% [23]. In addition, in the case of fucoidan from *S. hemiphyllum* having 23% sulfate groups, even at a concentration of 1 mg/mL, cell viability was found to be close to that of the control group, and it was confirmed that there was no inhibitory effect on Caco-2 [36]. Furthermore, whereas fucoidan isolated from *S. cinereum* increased ROS in Caco-2 cells in a dose-dependent manner [35], HMWF treatment reduced ROS levels which were elevated by MG-H1 treatment in Caco-2 cells. According to recent research, fucoidan can alleviate chronic colitis by preserving the intestinal mucosa and reducing crypt impairment in the colon of dextran sodium sulfate (DSS)-treated mice [37]. Also, fucoidan enhance intestinal integrity by inhibiting H_2_O_2_-induced oxidative stress and enhancing claudin-1 expression in Caco-2 cells [38]. Dietary fucoidan can enhance immune functions through immunomodulatory and anti-inflammatory effects [39]. Among fucoidan extracted from various brown seaweeds, fucoidan from *U. pinnatifida* possesses a higher sulfate level and exhibited the antioxidant capacity which is enhanced in proportion to the sulfate content [17]. According to our previous study, sulfate content of HMWF is 30.9%, which is higher than that of the low molecular weight fraction of fucoidan (LMWF). Other investigations discovered that low molecular weight fucoidan had more biological activity. Indeed, *S. hemiphyllum*’s low molecular weight fucoidan (0.8 kDa) improves intestinal barrier integrity and immunological function [36]. However, in a pharmacokinetic study, the mean residence time of low molecular weight in blood (109 min) is lower than high molecular weight fucoidan in blood (6.79 h), indicating that the latter circulates for a longer period of time. As a result, the high molecular fucoidan is slowly eliminated from circulation and might reside in various organs. Furthermore, as compared to LMWF at the same concentration, the HMWF treatment considerably increased the proliferation of NK cells, indicating that HMWF has a greater immune stimulatory effect [23]. In addition, the mean residence time of high molecular weight is 14.57 h, and daily oral treatment of the high molecular weight fucoidan for 4 weeks can reverse cyclophosphamide-induced immunosuppression in the mouse spleen [40]. Thus, we investigated the protective effects of HMWF against intestinal barrier dysfunction in this study.

The intestinal epithelial cells serve as a primary pathway related to the influx of foods and tight barricade for harmful compounds such as toxicants derived from diet or bacteria [41,42]. The vast surface encounters dAGEs, which principally come from the diet. Permeabilization of intestinal barrier contributes the translocation of dAGEs as well as other toxic materials including LPS from gut lumen to systemic circulation, activating immune responses and inflammation [29]. Moreover, many studies have reported that a disruptive intestinal barrier function is not only found in intestinal diseases but is also associated to other metabolic disorders such as hepatic disorders (e.g., non-alcoholic fatter liver disease), and renal failure [43,44]. Long exposure to high dAGEs significantly increased colon permeability in rat colon through reducing TJ protein expression of occludin and ZO-1 [45]. However, the impact of MG-H1 on intestinal integrity has not been thoroughly studied. Without binding to the protein carrier, single dAGEs such as MG-H1 are accumulated in intestinal epithelial cells because they are strongly trapped inside the cells [46]. Our results showed that MG-H1 treatment reduced TEER value and increased the permeability coefficient in Caco-2 cell monolayers. Besides, the mRNA and protein expressions of TJ components including ZO-1, occludin and claudin-1 are diminished in Caco-2 cells and colon tissues. As a result, MG-H1 might disrupt intestinal barrier function. On the other hand, HMWF administration reverses the effect of MG-H1 on intestinal integrity in both in vitro and in vivo. It should be noted that the Papp value was estimated indirectly using the FITC-dextran flux rather than directly using the fucoidan-FITC method. The Papp was 1.5 ± 0.05 × 10^−6^ cm/s when HWWF alone was treated in the cells (data not shown), which was close to that of the control group (1.32 ± 0.03). In addition, according to a previous study, the results of direct fucoidan absorption and efflux Papp through Caco-2 cells suggested fucoidan can be absorbed [47]. Furthermore, TJ marker levels increased which may have a positive effect on barrier integrity. HMWF treatment inhibited MG-H1-induced reduction of TJ marker expression in Caco-2 cells and mouse colon tissues, including zonula occludens-1, occludin, and claudin-1 at the mRNA and protein levels. However, because many other claudins, junctional adhesion molecules, and tricellulin were not evaluated in this work, further research into the expression of these markers is required to assess intestinal permeability.

Oxidative stress is the one of the main causes of an epithelial barrier dysfunction [48]. Excessive generation of ROS and its accumulation to tissues in the gastrointestinal tract can cause DNA damage in the mitochondria of epithelial cells leading to cellular damage and subsequent intestinal barrier impairment [49]. Fucoidan from *U. pinnatifida* whose molecular weight is larger than 300 kDa exerts significant secondary antioxidant activity and known to be a scavenger of ROS. In lined with this, HMWF treatment significantly decreased generation of ROS. On the other hand, MG-H1 increased cellular ROS production on this study. Opposite to the well-characterized dAGEs such as CML and CEL, MG-H1 does not need a peptide carrier for binding to RAGE to exert its effect [13]. In addition, MG-H1 content of the processed foods is detected at the highest level. Therefore, free MG-H1 absorbed from the processed foods is most likely a principal activator of RAGE [11,50]. RAGE activation through AGE/RAGE axis in the GI tract may promote intestinal inflammation by stimulating localized production of ROS and pro-inflammatory cytokines, both of which have been shown to impair TJ between epithelial cells, eventually injuring gut barrier integrity and contributing to the pathogenesis of IBD [12]. Thus, we speculated that the interaction between MG-H1 and RAGE is associated with the induction of ROS and inflammation. Our results showed that MG-H1 significantly increased RAGE in Caco-2 cells and mouse colon tissues. Also, MG-H1 increased the mRNA expression of TNF-α. TNF-α causes disruption of TJ proteins ZO-1 in differentiated Caco-2 monolayers [51]. In addition, MPO activity was increased by MG-H1 treatment in mouse colon tissues. Because MPO is a neutrophil-specific enzyme that acts as a marker of neutrophil infiltration and acute inflammation, MG-H1-induced MPO activity implies acute inflammation in the mouse colon [52]. Mizumoto et al. reported that sulfated glycosaminoglycans such as chondroitin sulfate and heparin sulfate strongly bind to RAGE and inhibit its effects [53]. Furthermore, as a highly sulfated glycosaminoglycan, fucoidan and heparin share a similar molecular structure [54]. As a result, we expect that HMWF significantly decreased the expression of RAGE and TNF-α, indicating that HMWF protected intestinal TJ against inflammation induced by interaction between MG-H1 and RAGE.

As a major driver of inflammation, the activation of NF-κB is closely related to up-regulation of RAGE in inflamed intestinal tissues gathered from IBD patients [55]. Increased oxidative stress induced by RAGE activation leads to continuous activation of NF-κB, with succeeding generation of ROS and overexpression of pro-inflammatory mediators such as TNF-α, COX-2, and iNOS [56]. TNF-α is a well-known intestinal integrity-disrupting agent involved in the occurrence of inflammatory colitis, secreted by macrophages and monocytes [57]. COX-2 and iNOS are important pro-inflammatory mediators, which play critical roles in the progress of intestinal damage [58,59]. Our results figured out that MG-H1 treatment increased expression of COX-2 and iNOS at protein level. Furthermore, MG-H1 treatment increased the level of NF-κB expression. NF-κB could regulate the expression of COX-2 and iNOS [60,61]. In accordance with our study, free CML increased the expression of RAGE and NF-κB indicating increased AGEs-induced oxidative stress [62]. It has been found that anti-inflammatory effect of fucoidan is derived from the attenuation of the NF-κB signaling pathway [16,63]. Also, HMWF from *L. japonica*, whose sulfate content is similar to our HMWF, decreased production of COX-2 and iNOS by regulating the phosphorylation of NF-κB [64]. In line with earlier research, HMWF from *U. pinnatifida* inhibited NF-κB phosphorylation and lowered COX-2 and iNOS production, demonstrating the anti-inflammatory properties of HMWF.

## 4. Materials and Methods

### 4.1. Materials

HMWF were derived from *U. pinnatifida* and prepared by Haerim Fucoidan Co., Ltd. (Wando, Republic of Korea). HMWF’s chemical composition was originally reported to contain fucose (21%), galactose (23%), mannose (0.9%), uronic acid (10.9%), sulfate (30.9%), and a peak molecular weight of 258.7 kDa as determined by high performance gel permeation chromatography [23]. For the cell culture, Dulbecco Modified Eagle Medium (DMEM) medium was purchased from GIBCO (Brooklyn, NY, USA) and fetal bovine serum, trypsin-EDTA, and penicillin-streptomycin were purchased from Hyclone (Logan, UT, USA). MTT was purchased from Sigma-Aldrich (St. Louis, MO, USA). Monoclonal primary antibodies, which include anti-claudin-1, anti-occludin, anti-RAGE, anti-COX-2, anti-iNOS, anti-NF-κB anti-p-NF-κB and anti-GAPDH, as well as secondary antibodies such as anti-mouse, anti-rabbit and anti-rat, were purchased from Santa Cruz Biotechnology Inc. (Santa Cruz, CA, USA). Anti-ZO-1 antibody was purchased from Cell Signaling Technology Inc. (Denvers, MA, USA). MG-H1 was purchased from Iris Biotech (Marktredwitz, Germany).

### 4.2. Treatment and Experimental Design

Male ICR mice, seven weeks old, were purchased from Orientbio, Inc. (Seongnam, Republic of Korea). All experimental mice were housed in a environment at 20 ± 3 °C, relative humidity 50 ± 10% with 12 h light/dark cycle, and had free access to standard diet and tap water. This study was conducted under the directions of the Committee for Ethical Usage of Experimental Animals of Korea University (KUIACUC-2021-0105). Mice were acclimatized for one week and the divided into four groups (8 mice/group): control group, MG-H1 treatment group, HMWF low treatment group and HMWF high treatment group. For 4 weeks, 400 μg/kg b.w./day of MG-H1 was intravenously administered through injection at tail vein and 25 or 75 mg/kg b.w./day of HMWF was orally administered.

### 4.3. Histological Analysis and Intestinal Permeability Analysis

After 4 weeks, 32 mice were sacrificed, and colon samples were taken from each group. Removed colon tissues were fixed with 10% neutral formalin to perform histological analysis. The sections were deparaffinized and stained with hematoxylin and eosin.

Intestinal permeability was evaluated by using FITC-dextran as previously described with some modifications [65]. The FITC-dextran powder was dissolved in phosphate-buffered saline to 50 mg/mL. Briefly, mice were fasted for 12 h and then orally gavaged with 200 mg/kg b.w. FITC-dextran solution. After 4 h, a venous blood sample was taken, and serum was isolated by centrifugation 3000× *g* for 10 min at 4 °C. The FITC-dextran that crossed from the lumen into blood was measured by a fluorometer (HIDEX, Turku, Finland) at wavelength (λ_ex_ = 490 nm and λ_em_ = 520 nm). The intestinal permeability was expressed as the amount of FITC-dextran exuded (micrograms per centimeter per min). A calibration curve for FITC-dextran was linear up to 0.1 mg/mL.

### 4.4. Measurement of MPO Activity

MPO activity was determined as previously described [66]. Weighed samples of colon tissues were homogenized in 0.5% hexadecyltrimethylammonium bromide buffer in 50 mM phosphate-buffered saline (pH 6) for 4 min at 30 Hz. Homogenates were centrifuged at 13,400× *g*, 6 min at 4 °C. Samples of supernatant were added to reaction mixture consisting of o-dianisidine hydrochloride and H_2_O_2_ (5 × 10^−4^%), and MPO activity was measured spectrophotometrically at 460 nm, and expressed as units/g of tissue.

### 4.5. Cell Culture and Viability Assay

Caco-2 cell line (ATCC, Manassas, VA, USA) (passage number 20–35) was maintained on a Dulbecco’s Modified Eagle’s Medium (DMEM; low glucose, Gibco, NY, USA) containing sodium bicarbonate of 3.7 g/L and 1% (*v*/*v*) penicillin/streptomycin with 1% (*v*/*v*) non-essential amino acid solution in Caco-2. The cells were incubated in humidified incubator at 37 °C and 5% CO_2_ concentration.

To measure cell viability, we used MTT assay. The Caco-2 cells were seeded on 24-well at a density of 1 × 10^5^ cells/well and incubated for 24 h, followed by treatment with MG-H1 and HMWF for 24 h. The MTT solution was then added to the well and left to sit for 3 h. Next, the MTT solution was removed, and DMSO was added to dissolve the insoluble formazan. Finally, the absorbance was measured using a microplate reader (EL-808, BioTek, Bad Friedrichshall, Germany) at 540 nm.

### 4.6. Measurement of Intracellular ROS

Cellular ROS were quantified by the DCFH-DA assay using a microplate reader as previously described [67]. Cells were cultured in 96-well plate at a density of 2 × 10^4^ cells/well and changed to fetal bovine serum-free medium before the assay. Each sample was subjected to six parallel experiments. At the end of the incubation time, the culture media was aspirated, and 20 μM DCFH-DA was added to each well and incubated at 37 °C for 30 min, during which oxidation of dichlorodihydrofluorescein, DCFH (no fluorescence) by ROS changed the molecule to dichlorofluorescein, DCF (fluorescence). After the incubation, the wells were washed twice with sterile phosphate-buffered saline, and measured by a fluorometer (HIDEX, Turku, Finland) at wavelength (λ_ex_ = 485 nm and λ_em_ = 530 nm).

### 4.7. Transepithelial Electrical Resistance (TEER) and Permeability Assays

The Caco-2 cells were seeded at a density of 1 × 10^5^ cells/well in 12 Transwell™ inserts (Polyethylene terephthalate membranes, 0.4 μm pore size, 0.9 cm^2^ growth area; Falcon, NY, USA) and cultured for 21 days to allow complete cell differentiation and monolayer formation. During the differentiation, the culture medium was replaced every other day. Before measuring the TEER with an ohm/voltmeter (EVOM, WPL, Sarasota, FL, USA), we replaced the medium with Hank’s balanced salt solution (HBSS) and added sodium bicarbonate. The following formula is used to calculate TEER values:
$$\mathrm{TEER}\;(Ω\cdot {\mathrm{cm}}^{2})=(R_{\mathrm{total}}\;-\;R_{\mathrm{blank}})\;\times \;\mathrm{growth}\;\mathrm{area}\;({\mathrm{cm}}^{2})$$
where R_total_ is the resistance across the cell layer on the semipermeable membrane; R_blank_ is the blank resistance of the semipermeable membrane only (without cells). The integrity of the cell monolayer was routinely checked before MG-H1 and HMWF treatments by measuring the TEER value. TEER levels were evaluated following the end of the treatments, and the results were represented as a percentage of the final value in comparison to the control group. Differentiated Caco-2 cell monolayers with TEER > 200 Ω·cm^2^ were used for the TEER and permeability experiments.

To evaluate paracellular permeability, we measured the flux of fluorescein isothiocyanate (FITC)-dextran 4 kDa (46944, Sigma, St. Louis, MO, USA) in Caco-2 monolayer. After treatment, 1 mg/mL FITC-dextran was added to apical side and incubated for 2 h at 37 °C. Samples were collected from the basolateral side, and the fluorescence was measured using a fluorometer (HIDEX, Turku, Finland) at wavelength (λ_ex_ = 490 nm and λ_em_ = 520 nm). FITC-dextran flux is expressed as permeability coefficient (Papp), which was calculated by the following formula:$$\mathrm{Papp}\;(\mathrm{cm}/s)=(\frac{V_{A}}{A\times t})\times \left(\frac{C_{f}}{C_{i}}\right)$$
where Papp is permeability coefficient: *V_A_* and *C_f_* are volume (cm^3^) and concentration (μM) of receiver (basolateral side), respectively; *A* is membrane growth area (cm^2^); *t* is assay time (s); and *C_i_* is initial apical concentration (μM).

### 4.8. Quantitative Real-Time PCR Analysis

Isolation of total RNA was performed by RNAiso PLUS (Takara, Kusatsu, Japan). The RNA concentration was measured using NanoDrop™2000 (Thermo Scientific, IL, USA). First-strand cDNA was synthesized in accordance with the instructions of a Premium Express 1st strand cDNA synthesis System (Legene Biosciences, San Diego, CA, USA). Quantitative real-time PCR reactions were performed with SYBR green (ELPIS Biotech. Inc., Daejeon, Republic of Korea) in CFX96™ Real-Time PCR (Bio-Rad Laboratories, Inc., Hercules, CA, USA). The human and mouse qRT-PCR primer sequences were shown in Appendix A. GAPDH was calibrated using the housekeeping gene, and the data were measured using the 2^−∆∆Ct^ method.

### 4.9. Western Blot Analysis

Mouse colon tissue and Caco-2 cells were lysed using a radioimmunoprecipitation (RIPA) buffer which consisted of 50 mM NaCl/50 mM Tris-HCl (pH 7.5), 0.2% deoxycholic acid/0.5% Triton X-100™/1% Nonidet P-40 containing 0.1%, and 1 mM phenylmethylsulfonyl fluoride, in the presence of aprotinin and leupeptin. The supernatant obtained using centrifugation (7558× *g*, 20 min) was utilized as a total cell lysate. Protein samples were prepared using the BCA assay and separated by SDS-PAGE and transferred to a polyvinylidene membrane (Millipore, Billerica, MA, USA). The membrane was blocked by using 5% skim milk for 45 min. Then, the membrane was incubated overnight in a solution containing the primary antibody at 4 °C and subsequently washed and incubated for 1 h with the secondary antibody at 25 °C. Finally, the protein was detected by an Amersham ECL select™ reagent kit (RPN2235). The protein band signal was quantified using Image J software (National Institutes of Health, Bethesda, MD, USA).

### 4.10. Immunofluorescence

Localization of the TJ proteins was evaluated using immunofluorescent staining in differentiated Caco-2 cells. The cells were seeded onto a 13-mm confocal dish (200350, SPL Life Sciences CO., Ltd., Korea) at a density (1 × 10^5^ cells/well). The differentiated Caco-2 cells were treated with MG-H1 and HMWF. Next, 4% paraformaldehyde was added to wells, and then 0.1% triton X-100 was added to wells for fixation and permeabilization. After that, it was treated for 1 h with 1% bovine serum albumin to block the cells. Next, the cells were incubated with primary antibody at 4 °C overnight, followed by incubation with secondary (Abcam Inc., Cambridge, CA, USA) for 2 h. Nuclei were stained using DAPI solution of 500 ng/mL for 10 min. The localization of occludin was observed by a confocal laser scanning microscope (CLSM 700; Carl Zeiss, BA, Germany).

### 4.11. Statistical Analysis

All statistical analyses were performed using the SAS version 9.4 (SAS institute, Cary, NC, USA). Data were expressed as mean ± SD. All statistical comparisons were made by one-way ANOVA followed by Tukey’s multiple range test and Dunnett’s *t*-test. The differences between groups with *p*-values < 0.05 were considered statistically significant.

## 5. Conclusions

The current study demonstrated that the HMWF treatment alleviated intestinal barrier dysfunction in Caco-2 cells via modulating MG-H1-induced intracellular ROS production and TJ loss. HMWF administration restored TJ expression and decreased intestinal inflammation in MG-H1-treated mice colon tissues through modulating inflammatory mediator expression and MPO activity. HMWF treatment reduced intestinal barrier dysfunction caused by MG-H1/RAGE axis-induced inflammation via NF-B signaling. As a result, HMWF derived from *U. pinnatifida* might be utilized as a dietary supplement to improve gut health.

## Figures and Tables

**Figure 1 marinedrugs-20-00580-f001:**
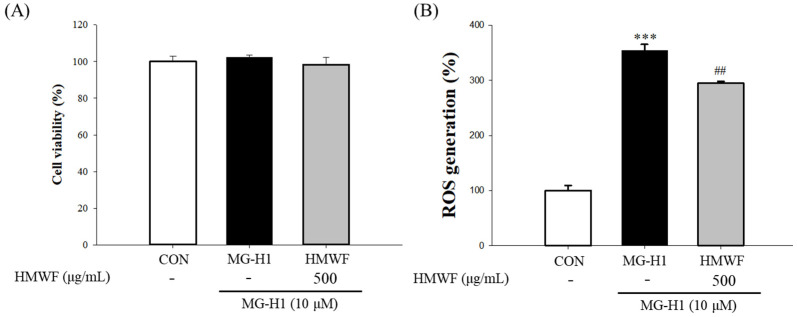
Effect of high molecular weight fucoidan (HMWF) on methylglyoxal-derived hydroimidazolone-1 (MG-H1) induced cell cytotoxicity of the human colon carcinoma cell line (Caco-2) cells. (**A**) Cells were treated with 500 μg/mL HMWF and 10 μM MG-H1 for 24 h. Caco-2 cell viability was evaluated using 3-[4,5-dimethylthiazol-2-yl]-2,5-diphenyltetrazolium bromide (MTT) assay. (**B**) Cells were treated with 500 μg/mL HMWF and 10 μM MG-H1 for 24 h. The reactive oxygen species (ROS) level was evaluated by 2,7-dichlorofluorescin diacetate assay. The results are shown as mean ± standard deviation (*n* = 3). *** *p* < 0.001 versus the control cells, ^##^ *p* < 0.01 versus MG-H1 treated cells.

**Figure 2 marinedrugs-20-00580-f002:**
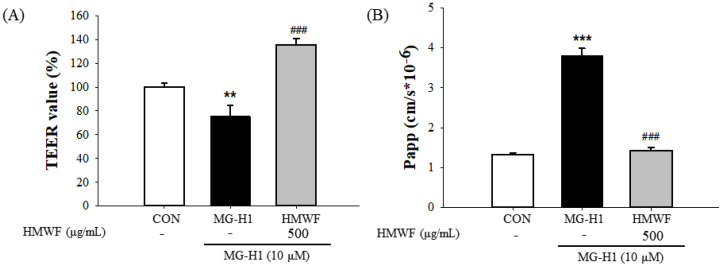
The effect of HMWF on monolayer integrity of differentiated Caco-2 cells. Caco-2 cell monolayers were incubated with 500 μg/mL HMWF and 10 μM MG-H1. (**A**) Caco-2 cell monolayer integrity was evaluated by measuring the transepithelial electrical resistance (TEER). (**B**) The permeability coefficient (Papp) was determined with the flux of fluorescein isothiocyanate. The results are shown as mean ± standard deviation (*n* = 3). ** *p* < 0.01, *** *p* < 0.001 versus the control cells, ^###^ *p* < 0.001 versus MG-H1 treated cells.

**Figure 3 marinedrugs-20-00580-f003:**
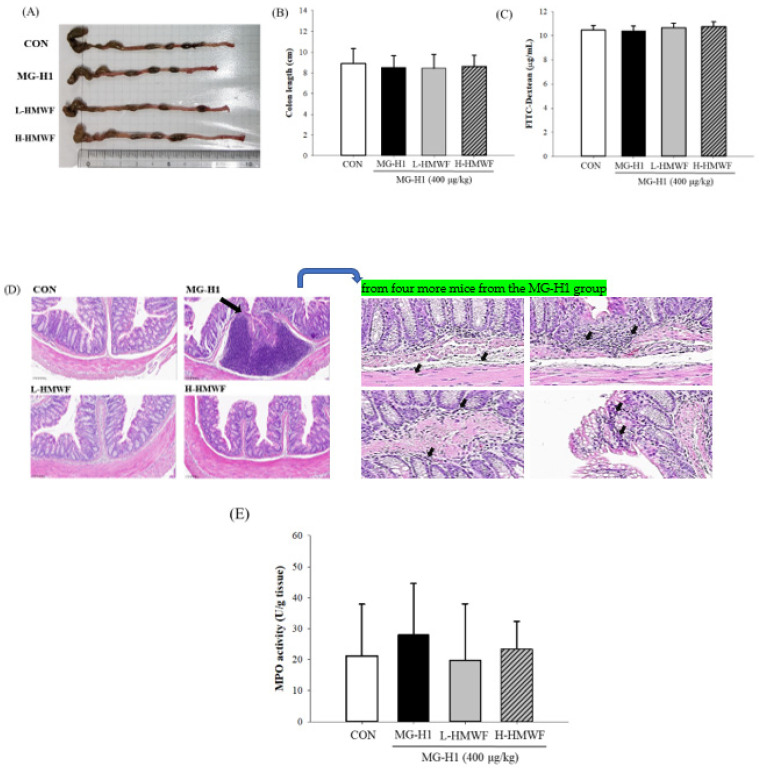
Effect of HMWF on colon damage in intravenously MG-H1-treated mice. For four weeks, mice were given high dose of HMWF (75 mg/kg body weight (b.w.)) or low dose (25 mg/kg b.w.) orally and MG-H1 (400 μg/kg b.w.) intravenously once a day. (**A**) Photographs of the colon. (**B**) colon length. (**C**) Serum fluorescein isothiocyanate (FITC) levels. (**D**) A representative hematoxylin and eosin (H&E) staining of the colon tissues for each group (20×). Neutrophil infiltration (black arrow) from H&E-stained samples from four more mice from the MG-H1 group can be seen on the right. (**E**) Myeloperoxidase (MPO) activity. All values are presented as the mean ± SD (*n* = 8). Control; control group, MG-H1; Intravenous MG-H1 injection group, L-HMWF; Group of intravenous MG-H1 injection + low dose of HMWF (25 mg/kg b.w.) oral administration, H-HMWF; Group of intravenous MG-H1 injection + high dose of HMWF (75 mg/kg b.w.) oral administration. The data were analyzed by Dunnett’s *t*-test.

**Figure 4 marinedrugs-20-00580-f004:**
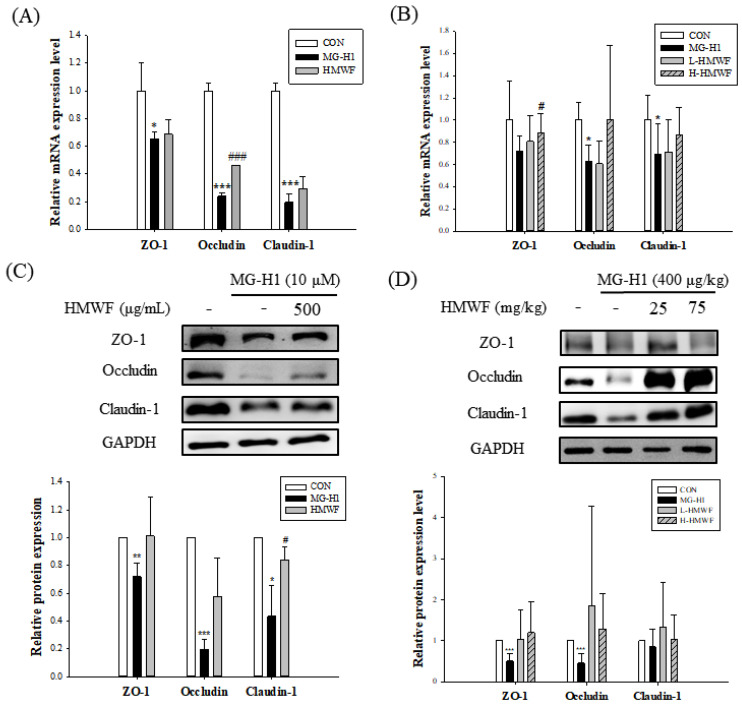
Effect of HMWF on expression of tight junction (TJ) markers. Caco-2 cell monolayers were treated for 24 h with 500 μg/mL HMWF and 10 μM MG-H1 for 24 h. Mice were treated with 25 or 75 mg/kg b.w. HMWF orally and 400 μg/kg b.w. MG-H1 intravenously once a day for 4 weeks. (**A**) mRNA expression of TJ markers in Caco-2 monolayers and (**B**) mouse colon tissues. (**C**) Protein expression of TJ markers in Caco-2 monolayers and (**D**) mouse colon tissues. The results are shown as mean ± standard deviation. * *p* < 0.05 ** *p* < 0.01, *** *p* < 0.001 versus the control group, ^#^ *p* < 0.05, ^###^ *p* < 0.001 versus the MG-H1-treated group.

**Figure 5 marinedrugs-20-00580-f005:**
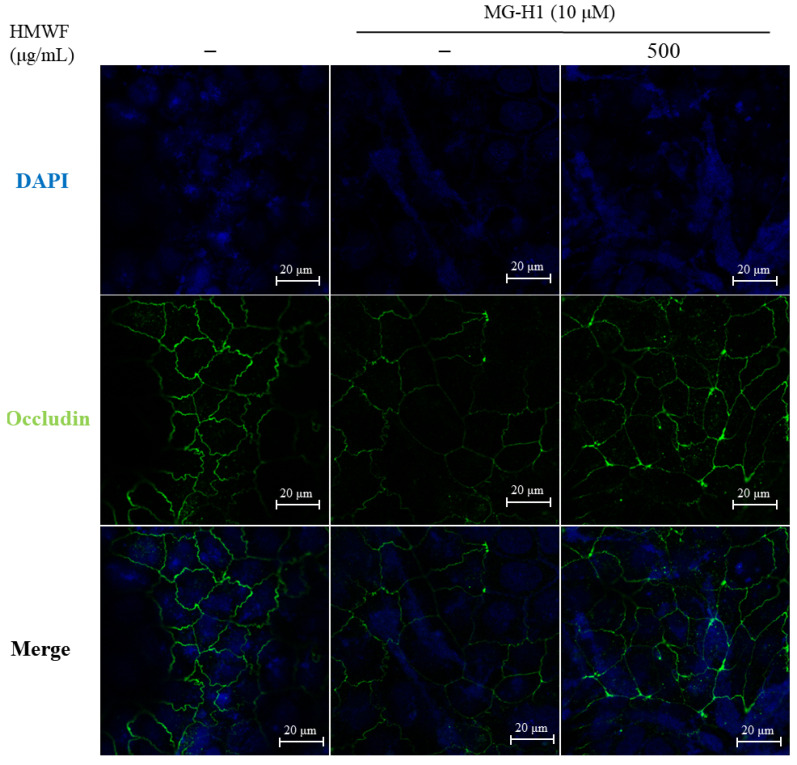
Effect of HMWF on localization of occludin in Caco-2 cells. Localization of occludin was measured using immunofluorescence in Caco-2 cells. The cells were differentiated for 5 d and then treated with 500 μg/mL HMWF and 10 μM MG-H1 for 24 h. The localization of occludin (green) and nucleus (DAPI, blue) in intercellular was measured using 20× objective by confocal laser scanning microscope with specific antibody; scale bar 20 μm.

**Figure 6 marinedrugs-20-00580-f006:**
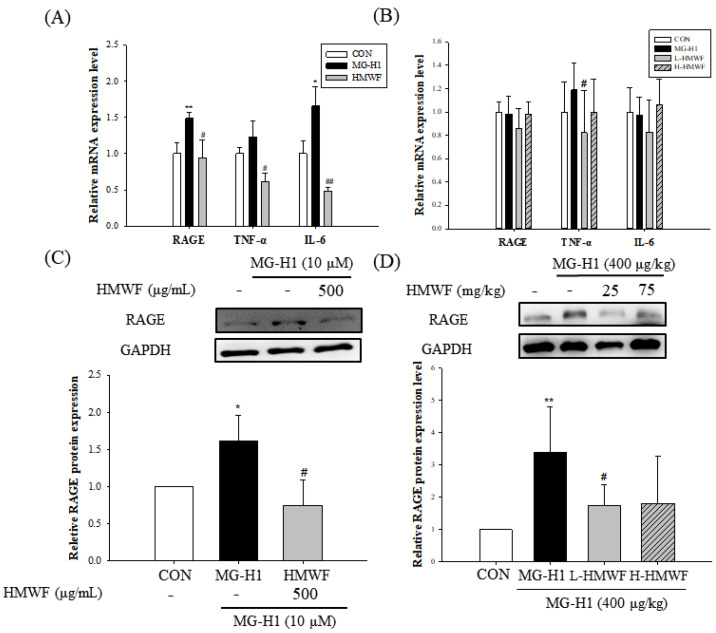
Effect of HMWF on RAGE-induced intestinal inflammation. Caco-2 cell monolayers were incubated with 500 μg/mL HMWF and 10 μM MG-H1 for 24 h. Mice were treated with HMWF (25 or 75 mg/kg b.w.) orally and MG-H1 (400 μg/kg b.w.) intravenously once a day for 4 weeks. (**A**) The mRNA expression of RAGE, TNF-α, and IL-6 in Caco-2 monolayers and (**B**) mouse colon tissue. (**C**) Protein expression of RAGE in Caco-2 monolayers and (**D**) mouse colon tissue. The bar graph of the relative intensities of western blotting bands. All values are presented as the mean ± SD. The data were analyzed by Dunnett’s *t*-test. * *p* < 0.05, ** *p* < 0.01 versus the control group, ^#^ *p* < 0.05, ^##^ *p* < 0.01 versus the MG-H1-treated group.

**Figure 7 marinedrugs-20-00580-f007:**
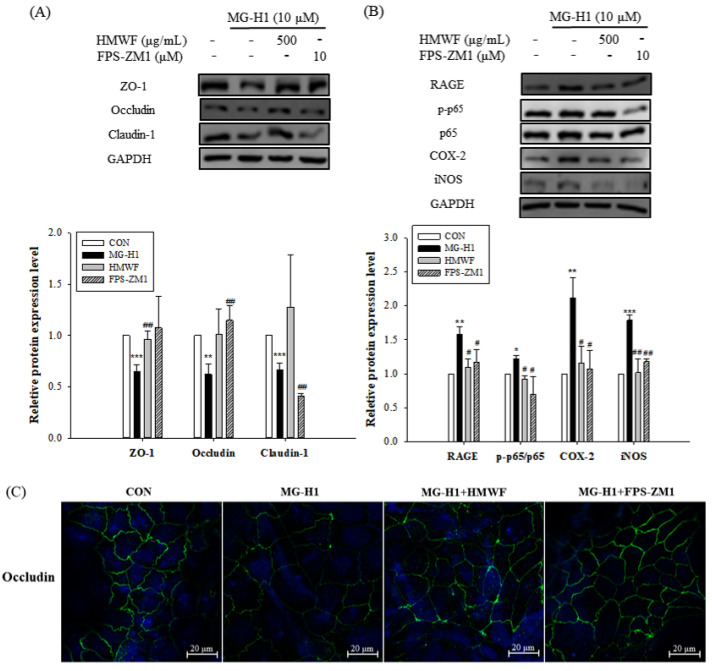
Effect of HMWF on RAGE-NF-κB signaling in intestinal inflammation in Caco-2 cells. Caco-2 cell monolayers were pre-treated with 10 μM FPS-ZM1, a selective RAGE inhibitor before incubation with 10 μM MG-H1. (**A**) The protein expression of TJ markers in Caco-2 cells. (**B**) The protein expression of inflammatory mediators involved in RAGE signaling pathway in Caco-2 cells. The bar graph of the relative intensities of western blotting bands. (**C**) Localization of occludin measured by immunofluorescence. All values are presented as the mean ± SD. The data were analyzed by Dunnett’s *t*-test. * *p* < 0.05 versus the control cells, ** *p* < 0.01 versus the control cells, *** *p* < 0.001 versus the control cells, ^#^ *p* < 0.05, ^##^ *p* < 0.01 versus the MG-H1-treated cells; The localization of occludin (green) and nucleus (DAPI, blue) was determined using confocal laser scanning microscope with specific antibody; original magnification 20×; scale bar 20 μm.

## Data Availability

The data that support the findings of this study are available on request from the corresponding author.

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
