# Peer review of "High Molecular Weight Fucoidan Restores Intestinal Integrity by Regulating Inflammation and Tight Junction Loss Induced by Methylglyoxal-Derived Hydroimidazolone-1"

_marinedrugs, 2022, doi:10.3390/md20090580_

Round 1

Reviewer 1 Report

Thank you for the opportunity to review this paper. In it, the authors have investigated the effects of high molecular weight fucoidans (HMWF) to mitigate the ROS generating effects of MG-H1 in two models – an in vitro cell culture model of intestinal epithelium and in vivo in a mouse model.

There are a number of concerns I have regarding the experimental setups used which affect the conclusions drawn form this study.

IN VITRO:

In the cell culture model, Caco2 cells were allowed to grow to confluence and treated with MG-H1 ± HMWF.  The dose that they chose to use for the HMWF was 500 ug/ml. This dose leads to an ~20% decrease in cell viability as assessed by MTT Assay (Fig 1A) (in presence of MG-H1, compared with MG-H1 alone). In the absence of MG-H1, HMWF at this dose significantly decreases cell viability (Fig S1). Thus, whether the effects seen in subsequent experiments are due to this dose causing a decrease in cell viability are unclear. It is unclear why the authors did not choose the highest dose which did not lead to a decrease in cell viability – i.e. 100 or 250 ug HMWF, in order to confidently rule out effects of reduced cell viability on being a factor here. Given the fact that the dosage used is affecting cell viability, we are unable to draw accurate conclusions from these data.

IN VIVO:

In The mouse model, male ICR mice were divided into groups and received

-          400 μg/kg b.w./day MG-H1 injected intravenously (no details are provided regarding the injection site)

-          And either 25 or 75 mg/kg 401 b.w./day of HMWF was orally administered for 4 weeks.

It is unclear why the authors elected to inject MG-H1 in this model. This leads to a systemic effect and whilst some of it may make its way to the intestinal epithelium, this is not a model of MG-H1 from exogenous sources e.g. diet (which the authors reference heavily – line 49-52, lines 267-272) Rather the injection of MG-H1 would better represent an increase in the endogenous production of AGEs. This experimental design does not permit conclusions to be made regarding the modulatory effect of HMWFs on exogenous (i.e. dietary) AGEs.

In the methods it is stated that mice were divided into groups of n=8 each, but only 5 were used for the analyses (Fig 3)? Why were not all animals included in the analysis?

Given there were no changes in a functional assessment of intestinal permeability (Dextran FITC) it is incorrect to say that based on mRNA and protein of several tight junction proteins that there was intestinal permeability. There are many different claudins, JAMs, tricellulin, etc which were not assessed. The current data does not provide convincing argument for increased intestinal permeabililty. It would be possible to assess in the plasma for markers of bacterial translocation, such as sCD14 or LBP.

MPO is significantly increased in colon tissues with MG-H1, which is reduced by HMWF. Visual assessment of H and E stained colon sections is not definitive for identification of neutrophils. Immunohistochemistry for neutrophil specific markers would be more appropriate.

Other issues:

-          It is unclear what purported mechanism the authors suggest fucoidan is working on.

-          Lines 130-134 try and sell that there was a difference (slight increase) however there are no statistical difference (possibly underpowered study if only n=5 mice included in analysis) – please tone down the language used here.

-          Line 84: Since ROS triggered by dietary AGEs can affect intestinal inflammation [24],  - this appears the wrong reference, as ref 24 is nothing to do with AGEs

-          Lines 36-39:  “In order to maintain an intact intestinal barrier, it is crucial for TJ proteins to interact and distribute  properly. Thus, changes in TJ proteins may result in modifications to intestinal integration.”  - This is unclear to the reader, could the authors elaborate on what they mean by “interact and distribute properly”

-          Line 46 – “Previous research have demonstrated systemic effects of dAGEs, but the localized effects 46 in the intestinal tract are not fully understood”

o   There has now been research which has shown the effects of dietary AGEs on the intestinal barrier – this should be acknowledge – PMID 33789895

-          Lines 267-272 – MG-H1 concentrations have been measured in blood samples – it would be better to cite these measured figures, rather than make estimates in this fashion, please refer to studies that have measured MG-H1 for assessing what would be expected PMID 26018950

-          Line 427 Caco cells – no details regarding passage numbers.

-          Line 434 No details regarding what equipment was used to measure TEER

-           

Statistical/Data Issues:

-          “Data availability statement: Not applicable.” – This data could be made available on a platform such as Open Science Framework. Given these are cell culture and animal experiments (as opposed to patients – where human ethics and privacy concerns may become apparent), there should be no issues with making the data available.

-          Many of the analyses involve 3 or more groups, yet the method says “The data were analyzed by Student’s t-test.” – please use correct statistical tests.

-          Data is presented as summary data (mean ± standard deviation). Please overlay individual data points to assist in reader interpretation of data.

Author Response

Reviewer(s)' Comments to Author:

Reviewer 1:

Comments and Suggestion for Authors

 Thank you for the opportunity to review this paper. In it, the authors have investigated the effects of high molecular weight fucoidans (HMWF) to mitigate the ROS generating effects of MG-H1 in two models – an in vitro cell culture model of intestinal epithelium and in vivo in a mouse model.

There are a number of concerns I have regarding the experimental setups used which affect the conclusions drawn from this study.

Comments:

1) IN VITRO:

In the cell culture model, Caco2 cells were allowed to grow to confluence and treated with MG-H1 ± HMWF.  The dose that they chose to use for the HMWF was 500 ug/ml. This dose leads to an ~20% decrease in cell viability as assessed by MTT Assay (Fig 1A) (in presence of MG-H1, compared with MG-H1 alone). In the absence of MG-H1, HMWF at this dose significantly decreases cell viability (Fig S1). Thus, whether the effects seen in subsequent experiments are due to this dose causing a decrease in cell viability are unclear. It is unclear why the authors did not choose the highest dose which did not lead to a decrease in cell viability – i.e. 100 or 250 ug HMWF, in order to confidently rule out effects of reduced cell viability on being a factor here. Given the fact that the dosage used is affecting cell viability, we are unable to draw accurate conclusions from these data.

à Answer:

First of all, we’d like to express our gratitude to the reviewer for the careful and critical reading of our manuscript.

With highlighted in green color text, we made the correction in response.

We repeated the cell viability experiment with cells treated with 500 mg/mL HMWF in the presence of MG-H1. As shown in Fig.1A, there was no significant reduction in cell viability in the control, MG-H1 and MG-H1 + HMWF group. Please see Fig. 1A. And the descriptions for the 500 mg/mL HMWF used in the study were given on page (P) 2, line (L) 87-89 as follows:

“As cell viability percentages above 80% are regarded non-toxic [24], and the highest absorption rate of fucoidan in Caco-2 cells was demonstrated at a concentration of 500 μg/mL [25]. Taken together, 500 μg/mL HMWF was employed in subsequent experiments.” 

2) IN VIVO:

In The mouse model, male ICR mice were divided into groups and received

-          400 μg/kg b.w./day MG-H1 injected intravenously (no details are provided regarding the injection site)

And either 25 or 75 mg/kg b.w./day of HMWF was orally administered for 4 weeks.

à We revised the statement on P13, L453-455 as follows:

“For 4 weeks, 400 μg/kg b.w./day of MG-H1 was intravenously administered through injection at tail vein and 25 or 75 mg/kg b.w./d of HMWF was orally administered.”

 It is unclear why the authors elected to inject MG-H1 in this model. This leads to a systemic effect and whilst some of it may make its way to the intestinal epithelium, this is not a model of MG-H1 from exogenous sources e.g. diet (which the authors reference heavily – line 49-52, lines 267-272) Rather the injection of MG-H1 would better represent an increase in the endogenous production of AGEs. This experimental design does not permit conclusions to be made regarding the modulatory effect of HMWFs on exogenous (i.e. dietary) AGEs.

à We appreciate your suggestion. The revised statements were added on P2, L51-56 as follows:

“Previous study has shown that consuming a heat-treated diet containing dAGEs for 24 weeks increases gut permeability, suggesting that dAGEs damage the intestinal barrier [8]. It has been reported that the Western diet, which has a relatively high proportion of sugar and fat, is linked to an increase in IBD incidences, hence increased intake of dAGEs is indicated [9]. Furthermore, dAGEs contribute to circulating AGEs and the AGEs pool in vivo, indicating that dAGEs can influence endogenous AGEs formation through AGEs metabolism [10].”

In the methods it is stated that mice were divided into groups of n=8 each, but only 5 were used for the analyses (Fig 3)? Why were not all animals included in the analysis?

à Please note that we have omitted the highest and lowest values for each group for the analysis. 

Given there were no changes in a functional assessment of intestinal permeability (Dextran FITC) it is incorrect to say that based on mRNA and protein of several tight junction proteins that there was intestinal permeability. There are many different claudins, JAMs, tricellulin, etc which were not assessed. The current data does not provide convincing argument for increased intestinal permeabililty. It would be possible to assess in the plasma for markers of bacterial translocation, such as sCD14 or LBP.

à First, in vitro results showed that MG-H1 significantly increased permeability in Caco-2 monolayer and decreased permeability by HMWF treatment. When conducting in vivo experiments, the DSS-induced colitis mice model, a well-known IBD model among intestinal-related diseases, was referred to. Although FITC Dextran had no change in in vivo permeability, when MG-H1 was delivered alone, the expression of tight junction-related factors (ZO-1 OCLN center) involved in paracellular permeability reduced. ZO-1 and OCLN expression increased again in the MG-H1+ HMWF group. We believed it was crucial to test in the circumstances of using a polymer like fucoidan because OCLN interacts to build a barrier that targets macromolecules. Additionally, ZO-1 is crucial for the reassembly of tight junctions because it binds OCLN and CLDN1 to the actin cytoskeleton. It also appears challenging to establish tricellulin's presence in the prior colon sample because it is known that this protein is primarily expressed in the small intestine of mice.

The concerned statements were added on P12, L374-379 as follows:

“Furthermore, TJ marker levels increased which may have a positive effect on barrier integrity. HMWF treatment inhibited MG-H1-induced reduction of TJ marker expression in Caco-2 cells and mouse colon tissues, including zonula occludens-1, occludin, and claudin-1 at the mRNA and protein levels. However, because many other claudins, junctional adhesion molecules, and tricellulin were not evaluated in this work, further research into the expression of these markers is required to assess intestinal permeability.”

 Visual assessment of H and E stained colon sections is not definitive for identification of neutrophils. Immunohistochemistry for neutrophil specific markers would be more appropriate.

à Answer: We provided the high-resolution image of neutrophil infiltration. Please see Fig. 3D.

3) Other issues:

-   It is unclear what purported mechanism the authors suggest fucoidan is working on.

à The concerned statements were added on P12, L331-332 as follows:

“Also, fucoidan enhance intestinal integrity by inhibiting H2O2-induced oxidative stress and enhancing claudin-1 expression in Caco-2 cells [38].”

In 1, L338-349:

“Other investigations discovered that low molecular weight fucoidan had more biological activity. Indeed, S. hemiphyllum's low molecular wight fucoidan (0.8 kDa) improves intestinal barrier integrity and immunological function [36]. However, in a pharmacokinetic study, the mean residence time of low molecular weight fucoidan (109 min) in blood is lower than high molecular weight one (6.79 h), indicating that the latter circulates for a longer period of time. As a result, the high molecular fucoidan is slowly eliminated from circulation and might reside in various organs. Furthermore, as compared to LMWF at the same concentration, the HMWF treatment considerably increased the proliferation of NK cells, indicating that HMWF has a greater immune stimulatory effect [23]. In addition, the mean residence time of high molecular weight fucoidan in the spleen is 14.57 h, and daily oral treatment of the high molecular weight one for 4 weeks can reverse cyclophosphamide-induced immunosuppression in the mouse spleen [40].”

In 12, L384-388:

“Fucoidan from U. pinnatifida whose molecular weight is larger than 300 kDa exerts significant secondary antioxidant activity and known to be a scavenger of ROS. In lined with this, HMWF treatment significantly decreased generation of ROS. On the other hand, MG-H1 increased cellular ROS production on this study.”

-  Lines 130-134 try and sell that there was a difference (slight increase) however there are no statistical difference (possibly underpowered study if only n=5 mice included in analysis) – please tone down the language used here.

à We modified the description on P4, L145-149 as follows:

“The amount of serum FITC-dextran is frequently used to estimate intestinal permeability. The serum concentration of FITC-dextran in the control group was 10.21 g/mL following oral administration of FITC-dextran, but a modest increase was seen in the MG-H1 treated group (10.63 g/mL). The blood level of FITC-dextran in the groups that received treatment with MG-H1+HMWF decreased somewhat but not significantly (Figure 3C).”

- Line 84: Since ROS triggered by dietary AGEs can affect intestinal inflammation [24],  - this appears the wrong reference, as ref 24 is nothing to do with AGEs

à The ref 24 was removed on P3, L93-94 as follows:

“Since ROS triggered by dAGEs can affect intestinal inflammation, 2,7-dichlorofluorescin diacetate ~”

- Lines 36-39:  “In order to maintain an intact intestinal barrier, it is crucial for TJ proteins to interact and distribute  properly. Thus, changes in TJ proteins may result in modifications to intestinal integration.”  - This is unclear to the reader, could the authors elaborate on what they mean by “interact and distribute properly”

à It was revised on P1-2, L40-44 as follows:

“Since the zonula occludens-1 (ZO-1), occludin, and claudin complexes in the intestinal epithelium are important for paracellular permeability, it is crucial for TJ proteins to interact and form tight connections in order to maintain an intact intestinal barrier [3].Thus, changes in TJ proteins may result in modifications to intestinal integration.”

- Line 46 – “Previous research have demonstrated systemic effects of dAGEs, but the localized effects 46 in the intestinal tract are not fully understood”

o   There has now been research which has shown the effects of dietary AGEs on the intestinal barrier – this should be acknowledge – PMID 33789895

à The study was cited on P2, L51-56 as follows:
“Previous study has shown that consuming a heat-treated diet containing dAGEs for 24 weeks increases gut permeability, suggesting that dAGEs damage the intestinal barrier [8]. It has been reported that the Western diet, which has a relatively high proportion of sugar and fat, is linked to an increase in IBD incidences, hence increased intake of dAGEs is indicated [9]. Furthermore, dAGEs contribute to circulating AGEs and the AGEs pool in vivo, indicating that dAGEs can influence endogenous AGEs formation through AGEs metabolism [10].”

- Lines 267-272 – MG-H1 concentrations have been measured in blood samples – it would be better to cite these measured figures, rather than make estimates in this fashion, please refer to studies that have measured MG-H1 for assessing what would be expected PMID 26018950

à We have revised on P10-11, L301-314 as follows:

“In addition to this, MG-H1 levels in blood whose eat processed foods and have insulin resistance ranges from 2.69 μM to 11.33 μM [33].”

-   Line 427 Caco cells – no details regarding passage numbers.

à The correction was made on P14, L481-482 as follows:

“Caco-2 cell line (ATCC, Manassas, VA, USA) (passage number 20-35) was maintained on a Dulbecco’s Modified Eagle’s Medium ~”

-  Line 434 No details regarding what equipment was used to measure TEER

à The revision was made on P15, L510-520 as follows”

“Before measuring the TEER with an ohm/voltmeter (EVOM, WPL, Sarasota, FL, USA), we replaced the medium with Hank's balanced salt solution (HBSS) and added sodium bicarbonate . The following formula is used to calculate TEER values:

TEER (Ω•cm2) = (Rtotal – Rblank) x growth area (cm2)

 Where Rtotal is the resistance across the cell layer on the semipermeable membrane; Rblank is the blank resistance of the semipermeable membrane only (without cells). The integrity of the cell monolayer was routinely checked before MG-H1 and HMWF treatments using measuring the TEER value. TEER levels were evaluated following the end of the treatments, and the results were represented as a percentage of the final value in comparison to the control group. Differentiated Caco-2 cell monolayers with TEER > 200 Ω•cm2 were used for the TEER and permeability experiments.”

 4) Statistical/Data Issues:

- “Data availability statement: Not applicable.” – This data could be made available on a platform such as Open Science Framework. Given these are cell culture and animal experiments (as opposed to patients – where human ethics and privacy concerns may become apparent), there should be no issues with making the data available.

à The correction was made on P16, L600-601 as follows:

“The data that support the findings of this study are available on request from the corresponding author.”

- Many of the analyses involve 3 or more groups, yet the method says “The data were analyzed by Student’s t-test.” – please use correct statistical tests.

à We correct it on P16, L574 as follows:

“Data were expressed as mean ± SD. All statistical comparisons were made by one-way ANOVA followed by Tukey’s multiple range test and Dunnett ’s t-test.”

-  Data is presented as summary data (mean ± standard deviation). Please overlay individual data points to assist in reader interpretation of data.

à We provided them as data (mean ± standard deviation) through the text.

Finally, we really appreciate the reviews’ critical and valuable comments which have been guided in revising our manuscript. We believe that our manuscript has been much improved due to the revision based on the reviewer’s suggestions.

Reviewer 2 Report

I have read the manuscript and I have questions and recommendations.
1. The composition and molecular weight of fucoidan is very important. Please provide the molecular weight, monosaccharide composition, sulfate and fucoidan content of your sample.
2. The bioavailability of fucoidan is related to its permeability. In your work, you made an indirect definition for the permeability coefficient. It is necessary to provide data for the permeability coefficient for the fucoidan itself. This is very important in terms of its application.

3. In the works (https://doi.org/10.3390/md16040132, https://doi.org/10.3390/md17120687) it was shown that high molecular weight fucoidan, both oral and topical, has good bioavailability. The work (https://doi.org/10.3402/fnr.v60.32033) also showed that fucoidan dramatically enhanced the intestinal epithelial barrier and immune function against the lipopolysaccharide effect by inhibiting IL-1β and TNF-α and promoting IL -10 and IFN-γ. Discuss the influence of molecular weight on the permeability and therefore the bioavailability of fucoidan.
4. In the work (https://doi.org/10.3390/ph15040418), various factors influencing the Caco-2 model in the study of fucoidan have been studied in detail. How did you select adequate conditions for a macromolecular compound in your work? Based on this, only two concentrations/doses were selected.
5. Fucoidan was previously found to inhibit the growth of Caco-2 cells in a dose-dependent manner (https://doi.org/10.1016/j.ijbiomac.2019.07.127). Compare your data with published data.
6. Correct, please, abstract. It should contain experimental data.
7. Specify the keywords, namely "advanced glycation end products" or give their chemical composition.

Author Response

Reviewer(s)' Comments to Author:

Reviewer 2:

Comments to the Author

I have read the manuscript and I have questions and recommendations.

à First of all, we’d like to express our gratitude to the reviewer for the careful and critical reading of our manuscript.

With highlighted in green color text, we made the correction in response.

1) The composition and molecular weight of fucoidan is very important. Please provide the molecular weight, monosaccharide composition, sulfate and fucoidan content of your sample.

à The information was provided on Page (P) 13, line (L) 430-434 as follows:

“HMWF were derived from U. pinnatifida and prepared by Haerim Fucoidan Co., Ltd (Wando, Republic of Korea). HMWF's chemical composition was originally reported to contain fucose (21%), galactose (23%), mannose (0.9%), uronic acid (10.9%), sulfate (30.9%), and a peak molecular weight of 258.7 kDa as determined by high performance gel permeation chromatography [23].”

2) The bioavailability of fucoidan is related to its permeability. In your work, you made an indirect definition for the permeability coefficient. It is necessary to provide data for the permeability coefficient for the fucoidan itself. This is very important in terms of its application.

à Answer: The concerned descriptions were added on P12, L369-374 as follows:

It should be noted that the Papp value was estimated indirectly using the FITC-dextran flux rather than directly using the fucoidan-FITC method. The Papp was 1.5 ± 0.05 x 10-6 cm/s when HWWF alone was treated in the cells (data not shown), which was close to that (1.32 ± 0.03) of the control group. In addition, according to a previous study, the results of direct fucoidan absorption and efflux Papp through Caco-2 cells suggested fucoidan can be absorbed [47].

3) In the works (https://doi.org/10.3390/md16040132, https://doi.org/10.3390/md17120687) it was shown that high molecular weight fucoidan, both oral and topical, has good bioavailability. The work (https://doi.org/10.3402/fnr.v60.32033) also showed that fucoidan dramatically enhanced the intestinal epithelial barrier and immune function against the lipopolysaccharide effect by inhibiting IL-1β and TNF-α and promoting IL -10 and IFN-γ. Discuss the influence of molecular weight on the permeability and therefore the bioavailability of fucoidan.

à Answer:

We appreciate the reviewer’s suggestion. The revision was made on P11, L338-349 as follows:

“Other investigations discovered that low molecular weight fucoidan had more biological activity. Indeed, S. hemiphyllum's low molecular wight fucoidan (0.8 kDa) improves intestinal barrier integrity and immunological function [36]. However, in a pharmacokinetic study, the mean residence time of low molecular weight fucoidan (109 min) in blood is lower than high molecular weight one (6.79 h), indicating that the latter circulates for a longer period of time. As a result, the high molecular fucoidan is slowly eliminated from circulation and might reside in various organs. Furthermore, as compared to LMWF at the same concentration, the HMWF treatment considerably increased the proliferation of NK cells, indicating that HMWF has a greater immune stimulatory effect [23]. In addition, the mean residence time of high molecular weight fucoidan in the spleen is 14.57 h, and daily oral treatment of the high molecular weight one for 4 weeks can reverse cyclophosphamide-induced immunosuppression in the mouse spleen [40].”

4)  In the work (https://doi.org/10.3390/ph15040418), various factors influencing the Caco-2 model in the study of fucoidan have been studied in detail. How did you select adequate conditions for a macromolecular compound in your work? Based on this, only two concentrations/doses were selected.

à Answer:

In the aforementioned study, it was determined if Caco-2 cells could differentiate when grown for 5 or 7 days throughout the course of 21 days under puromycin and fucoidan-containing conditions. The most important factor in Caco-2 differentiation is monolayer integrity since Caco-2 cells form a monolayer, thus that is where we'll start.

In terms of MTT assay, it was first demonstrated that there was no cytotoxicity up to a concentration of 500 mg/mL when the cells were treated with HMWF. Our preliminary experiment showed that the highest transport rate occurred at this concentration. Following 21 days of cell differentiation, the TEER assay and FITC flux were utilized to confirm the HMWF effectiveness in improving the barrier integrity impaired by MG-H1.

The descriptions were added on P2, L87-89 as follows:

“As cell viability percentages above 80% are regarded non-toxic [24], and the highest absorption rate of fucoidan in Caco-2 cells was demonstrated at a concentration of 500 μg/mL [25]. Taken together, 500 μg/mL HMWF was employed in subsequent experiments.”

On P4, L140-142 as follows:

“HWMF was given to the low dose (L-HMWF) and high dose (H-HMWF) in this investigation at doses of 25 and 75 mg/kg body weight (b.w.), respectively. In the previous experiment, both doses of HMWF treatments had immunostimulant effects [23].”

5)  Fucoidan was previously found to inhibit the growth of Caco-2 cells in a dose-dependent manner (https://doi.org/10.1016/j.ijbiomac.2019.07.127). Compare your data with published data.

à Answer:

We provided an explanation of the comparison between our data and published data on P11, L318-328 as follows:

“In the case of the MTT experiment, the cell viability decreased in a concentration-dependent manner from the concentration of 50 μg/mL for fucoidan from S. cinereum [35], but in the case of HMWF, the cell viability was maintained at more than 90% up to the concentration of 500 μg/mL. Fucoidan isolated from the brown algae Sargassum cinereum had a sulfate group concentration of 3.7% [35], whereas HMWF produced from Undaria pinnatifida had a sulfate group value of 30.9% [23]. In addition, in the case of fucoidan from S. hemiphyllum having 23% sulfate groups, even at a concentration of 1 mg/mL, cell viability was found to be close to that of the control group, and it was confirmed that there was no inhibitory effect on Caco-2 [36]. Furthermore, whereas fucoidan isolated from S. cinereum increased ROS in Caco-2 cells in a dose-dependent manner [35], HMWF treatment reduced ROS levels which were elevated by MG-H1 treatment in Caco-2 cells.”

6) Specify the keywords, namely "advanced glycation end products" or give their chemical composition.

à Answer:

We have specified the keywords on P1, L29-30 as follows:

Keywords: fucoidan; intestinal barrier; advanced glycation end product;  methylglyoxal-derived hydroimidazolone-1”

Finally, we really appreciate the reviews’ critical and valuable comments which have been guided in revising our manuscript. We believe that our manuscript has been much improved due to the revision based on the reviewer’s suggestions.

Reviewer 3 Report

The current research presented by Lim, et al. would investigate the remedy potential of high molecular weight fucoidan on intestinal inflammation. Several literature has proved the health benefits of fucoidans on bowel health, particularly for intestinal barrier functions as 10.3748/wjg.v19.i33.5500, 10.1371/journal.pone.0128453, and 10.3390/md19080436. In addition, the research is a further investigation of the authors (https://doi.org/10.3390/md17080447). However, the authors would unravel various molecular mechanisms mediating such effects, including the expression of pro-inflammatory cytokines and tight junction markers. However, there are drawbacks identified that hinder its publication in the present form. Such drawbacks can be classified into general and specific points. For instance,

General comments

-       Authors’ affiliation (No. 2) should be revised/omitted if they are not working in this institution.

-        The Abstract is too general. It should be thoroughly revised to show the applied experimental approaches and main findings by values.

-        References were missed in some places, e.g., line 34-39

-   The abbreviation meaning should be written completely with its first mention, e.g., ROS, Caco-2, MTT, MPO, etc., However, others as MG-H1 was written twice.

-        The space between the values and units should be unified, e.g., 24h and 24 h.   

-        A separate section for fucoidan potential chemistry and potential bioactivities is recommended to done.

-        A conclusion section should be separately discussed.

Specific comments

-        Despite of the previous authors’ publication (Ref. 22) discussing fucoidan chemical characterization, the downstream processes including fucoidan extraction and purification are not consistent with commonly used protocols using ethanol for precipitation and chromatography for purification. Also, three sugar standards were used for monomeric composition investigation. Nevertheless, other sugars as glucose should be involved which may indicate the contamination of crude fucoidan with other polysaccharides as laminarin. Additional references are needed to support this protocol of fucoidan isolation.

Hence, further chemical characterization using IR and specific fucoidan assays are recommended to be performed.

-        Which reference did the authors use to classify fucoidans into HMWF and LMWF using 300 MWCO?

-        May it be possible to incorporate a commercial anti-inflammatory drug in the current study for a comparison?

-        Comparison with low molecular weight fucoidan is also recommended.

-        Line 296: the difference in sulfate content between LMWF and HMWF is not so much potential to affect the bioactivity significantly (30.9% in HMWF vs. 28.8% in LMWF. Hence, further and adequate explanations are needed.   

-        Line 379: the word purified is not accurate!

-        Are there any references for the applied doses of fucoidan?

Unfortunately, the manuscript cannot be accepted in the present form, since a major revision is needed.

Author Response

Reviewer(s)' Comments to Author:

Reviewer 3:

Comments to the Author

The current research presented by Lim, et al. would investigate the remedy potential of high molecular weight fucoidan on intestinal inflammation. Several literature has proved the health benefits of fucoidans on bowel health, particularly for intestinal barrier functions as 10.3748/wjg.v19.i33.5500, 10.1371/journal.pone.0128453, and 10.3390/md19080436. In addition, the research is a further investigation of the authors (https://doi.org/10.3390/md17080447). However, the authors would unravel various molecular mechanisms mediating such effects, including the expression of pro-inflammatory cytokines and tight junction markers. However, there are drawbacks identified that hinder its publication in the present form. Such drawbacks can be classified into general and specific points. For instance,

à First of all, we’d like to express our gratitude to the reviewer for the careful and critical reading of our manuscript.

With highlighted in green color text, we made the correction in response.

1) General comments

 -        Authors’ affiliation (No. 2) should be revised/omitted if they are not working in this institution.

à Because the corresponding author is affiliated with two departments, the following was changed on page (P) 1, line (L) 5:

“Jae-Min Lim 1, Hee Joon Yoo 1 and Kwang-Won Lee 1,2*”

-        The Abstract is too general. It should be thoroughly revised to show the applied experimental approaches and main findings by values.

The abstract was revised according to the suggestion on P1, L12-28 as follows:

Abstract: Fucoidan from brown seaweeds has several biological effects, including preserving intestinal integrity. To investigate the intestinal protective properties of high molecular weight fucoidan (HMWF) from Undaria pinnatifida on intestinal integrity dysfunction caused by methylglyoxal-derived hydroimidazolone-1 (MG-H1), one of dietary advanced glycation end products (dAGEs) in the human colon carcinoma cell line (Caco-2) cells and ICR mice. According to research, dAGEs may damage the intestinal barrier by increasing gut permeability. The findings of the study showed that HMWF+MG-H1 treatment greatly reduced by 16.8% the amount of reactive oxygen species generated by MG-H1 treatment alone. Furthermore, HMWF+MGH-1 treatment reduced MG-H1-induced monolayer integrity disruption, as measured by alterations in transepithelial electrical resistance (135% vs. 75.5%) and fluorescein isothiocyanate incorporation (1.40 x 10-6 cm/s vs 3.80 cm/s). HMWF treatment prevented the MG-H1-induced expression of tight junction markers, including zonula occludens-1, occludin, and claudin-1 in Caco-2 cells and mouse colon tissues at the mRNA and protein level. Also, in Caco-2 and MG-H1 treated mice, HMWF plays an important role in preventing receptor for AGEs (RAGE)-mediated intestinal damage. In addition, HMWF inhibited the nuclear factor kappa B activation and its target genes leading to intestinal inflammation. These findings suggest that HMWF with price competitiveness could play an important role in preventing AGEs-induced intestinal barrier dysfunction.”

-        References were missed in some places, e.g., line 34-39

We added the refence on P1-2, L40-44 as follows:

“Since the zonula occludens-1 (ZO-1), occludin, and claudin complexes in the intestinal epithelium are important for paracellular permeability, it is crucial for TJ proteins to interact and form tight connections in order to maintain an intact intestinal barrier [3].Thus, changes in TJ proteins may result in modifications to intestinal integration.”

-   The abbreviation meaning should be written completely with its first mention, e.g., ROS, Caco-2, MTT, MPO, etc., However, others as MG-H1 was written twice.

à The correction were made according to the suggestion.

-        The space between the values and units should be unified, e.g., 24h and 24 h.   

à The correction were made according to the suggestion.

-        A separate section for fucoidan potential chemistry and potential bioactivities is recommended to done.

à Please note that we provide the statements on P2, L430-434 as follows:

“HMWF were derived from U. pinnatifida and prepared by Haerim Fucoidan Co., Ltd (Wando, Republic of Korea). HMWF's chemical composition was originally reported to contain fucose (21%), galactose (23%), mannose (0.9%), uronic acid (10.9%), sulfate (30.9%), and a peak molecular weight of 258.7 kDa as determined by high performance gel permeation chromatography [23].”

-        A conclusion section should be separately discussed.

à Answer: The correction was made on P16, L577-585 as follows:

 5. Conclusions

 The current study demonstrated that the HMWF treatment alleviated intestinal barrier dysfunction in Caco-2 cells via modulating MG-H1-induced intracellular ROS production and TJ loss. HMWF administration restored TJ expression and decreased intestinal inflammation in MG-H1-treated mice colon tissues through modulating inflammatory mediator expression and MPO activity. HMWF treatment reduced intestinal barrier dysfunction caused by MG-H1/RAGE axis-induced inflammation via NF-B signaling. As a result, HMWF derived from U. pinnatifida might be utilized as a dietary supplement to improve gut health.

 2) Specific comments

-        Despite of the previous authors’ publication (Ref. 22) discussing fucoidan chemical characterization, the downstream processes including fucoidan extraction and purification are not consistent with commonly used protocols using ethanol for precipitation and chromatography for purification. Also, three sugar standards were used for monomeric composition investigation. Nevertheless, other sugars as glucose should be involved which may indicate the contamination of crude fucoidan with other polysaccharides as laminarin. Additional references are needed to support this protocol of fucoidan isolation.

Hence, further chemical characterization using IR and specific fucoidan assays are recommended to be performed.

à The Haerim Fucoidan Company patented the downstream methods, including fucoidan extraction and purification, over which we have no influence. Unfortunately, please understand that we did not obtain further information from Haerim Fucoidan Co. on other polysaccharides, IR data, and fucoidan assays due to the company's stated restriction of delivering relevant information within time.

-        Which reference did the authors use to classify fucoidans into HMWF and LMWF using 300 MWCO?

à We referred to a study in which fucoidan from U. pinnatifida was fractionated based on 300 kDa MOCO to examine antioxidant properties.

Ref) Koh, H. S. A.; Lu, J.; Zhou, W., Structure characterization and antioxidant activity of fucoidan isolated from Undaria pinnatifida grown in New Zealand. Carbohydrate polymers 2019, 212, 178-185

-        May it be possible to incorporate a commercial anti-inflammatory drug in the current study for a comparison?

à We really appreciate the reviewer’s suggestion. Due to the due date, the supplementation was not made.

-        Comparison with low molecular weight fucoidan is also recommended.

à We added the statement on P11, L339-342 as follows:

“In addition, the mean residence time of high molecular weight fucoidan in the spleen is 14.57 h, and daily oral treatment of the high molecular weight one for 4 weeks can reverse cyclophosphamide-induced immunosuppression in the mouse spleen [40].”

-        Line 296: the difference in sulfate content between LMWF and HMWF is not so much potential to affect the bioactivity significantly (30.9% in HMWF vs. 28.8% in LMWF. Hence, further and adequate explanations are needed.   

à We added explanations on P11, L338-346 as follows:

“Other investigations discovered that low molecular weight fucoidan had more biological activity. Indeed, S. hemiphyllum's low molecular wight fucoidan (0.8 kDa) improves intestinal barrier integrity and immunological function [36]. However, in a pharmacokinetic study, the mean residence time of low molecular weight fucoidan (109 min) in blood is lower than high molecular weight one (6.79 h), indicating that the latter circulates for a longer period of time. As a result, the high molecular fucoidan is slowly eliminated from circulation and might reside in various organs. Furthermore, as compared to LMWF at the same concentration, the HMWF treatment considerably increased the proliferation of NK cells, indicating that HMWF has a greater immune stimulatory effect [23].”

-        Line 379: the word purified is not accurate!

The correction was made on P13, L430-431 as follows:

“HMWF were derived from U. pinnatifida and prepared by Haerim Fucoidan Co., Ltd (Wando, Republic of Korea).”

-        Are there any references for the applied doses of fucoidan?

à The revised descriptions were made on P2, L87-89 as follows:

“As cell viability percentages above 80% are regarded non-toxic [24], and the highest absorption rate of fucoidan in Caco-2 cells was demonstrated at a concentration of 500 μg/mL [25]. Taken together, 500 μg/mL HMWF was employed in subsequent experiments.”

on P4, L140-142 as follows:

“HWMF was given to the low dose (L-HMWF) and high dose (H-HMWF) in this investigation at doses of 25 and 75 mg/kg body weight (b.w.), respectively. In the previous experiment, both doses of HMWF treatments had immunostimulant effects [23].”

Finally, we really appreciate the reviews’ critical and valuable comments which have been guided in revising our manuscript. We believe that our manuscript has been much improved due to the revision based on the reviewer’s suggestions.

Round 2

Reviewer 1 Report

Thank the authors for considering my feedback and making many changes which will assist the reader in understanding the details of the experiments conducted in the manuscript. Thank you for including the necessary details in the methods sections, so that other scientists would be able to replicate these experimental conditions.

STATISTICAL CONCERNS RE: MOUSE STUDY

I am still concerned regarding the fact that only 5 mice from each group are used for the analysis, despite the fact that 8 were used for experiments. In the authors reply to my initial review, they state “we have omitted the highest and lowest values for each group for the analysis.”  This is not a valid statistical reason to remove 37.5% of your data from the analysis…if you are unsure as to why this is an issue, I recommend consulting with a statistical expert at your university. This may provide some insight into the issues regarding removing these data: https://scc.ms.unimelb.edu.au/resources/preparing-your-data/outliers

I recommend the authors include data from all mice in the manuscript.

NEUTROPHILS:

In response to my previous comments, the authors have including a higher resolution image of a H and E stain. However, the point I made previously was that H&E stain is not definitive for neutrophils. Given you have sections and slides, you can stain for a neutrophil specific marker (RB6-8C5 (anti-Gr-1) and then quantitate the amount of neutrophil accumulation. The current data presented is qualitative in nature (showing one image from one mouse) and doesn’t provide sufficient evidence to support the claims made on line 151.

Line 151: The MG-H1 group showed 151 the lymphoid aggregate in colon tissue which means infiltration of neutrophils. The lymphoid aggregates are effectively diminished in MG-H1+HMWF treated groups

All up, the manuscript has been improved with this recent revision. I hope these comments are useful and the authors are able to use this constructive feedback to improve the reporting of this research.

Author Response

Reviewer(s)' Comments to Author:

Reviewer 1:

Comments and Suggestion for Authors

 Comments:

1) STATISTICAL CONCERNS RE: MOUSE STUDY

I am still concerned regarding the fact that only 5 mice from each group are used for the analysis, despite the fact that 8 were used for experiments. In the authors reply to my initial review, they state “we have omitted the highest and lowest values for each group for the analysis.”  This is not a valid statistical reason to remove 37.5% of your data from the analysis…if you are unsure as to why this is an issue, I recommend consulting with a statistical expert at your university. This may provide some insight into the issues regarding removing these data: https://scc.ms.unimelb.edu.au/resources/preparing-your-data/outliers

I recommend the authors include data from all mice in the manuscript.

Answer:

First of all, again we’d like to express our sincere gratitude to the reviewer for the careful and critical reading of our manuscript.

With highlighted in green color text, we made the correction in response.

As suggested by the reviewer, we include data in which all values are presented as the mean ± SD (n=8) from all mice in the manuscript, the corrections were made on page (P) 4, lines (L) 142-147 as follows:

“The current results showed that the colon lengths among all groups showed no significant difference (Figure 3A-B). After oral administration of FITC-dextran, the serum concentration of FITC-dextran in the control group was 10.5 mg/mL; however, there was no significant difference in FITC-dextran levels between the MG-H1 treated group and the MG-H1+HMWF treated groups (Figure 3C).”

P4, L154-159:

“MG-H1 injection increased myeloperoxidase (MPO) activity (28.1±16.5 U/g tissue). Although there was no statistically significant difference from the MG-H1 treated group, HMWF treatment decreased the MG-H1-induced MPO activity (19.8±18.2 U/g tissue in L-HMWF group) and 23.4±8.9 U/g tissue in H-HMWF group) (Figure 3E). Thus, HMWF treatment may inhibit the MPO-mediated oxidative stress and colon inflammation induced by MG-H1.”

P6-7, L211-219:

“The mRNA expressions of TJ markers were significantly (0.63±0.14 and 0.69±0.27 in occludin and claudin-1 respectively) (p < 0.05) decreased using intravenous infusion of MG-H1 compared to the control (Figure 4B). The mRNA expression of ZO-1 was significantly (0.88±0.17) (p < 0.05) restored with H-HMWF administration (75 mg/kg b.w.). As shown in Figure 4D, in accordance with this tendency, treatment with MG-H1 significantly (p < 0.001)reduced the TJ markers' ZO-1 and occludin protein expression. Despite there being no significant difference, mice treated with HMWF treatment had higher expression of the proteins ZO-1 and occludin in their colons.”

P8, L242 and L254-255:

“~(1.74±0.65 and 1.80±0.47, respectively)~”

“~(1.18±0.22) (Figure 6B). HMWF oral treatment, on the other hand, significantly (0.82±0.36 in low concentration) (p < 0.05)~”

2) NEUTROPHILS:

In response to my previous comments, the authors have including a higher resolution image of a H and E stain. However, the point I made previously was that H&E stain is not definitive for neutrophils. Given you have sections and slides, you can stain for a neutrophil specific marker (RB6-8C5 (anti-Gr-1) and then quantitate the amount of neutrophil accumulation. The current data presented is qualitative in nature (showing one image from one mouse) and doesn’t provide sufficient evidence to support the claims made on line 151.

Line 151: The MG-H1 group showed 151 the lymphoid aggregate in colon tissue which means infiltration of neutrophils. The lymphoid aggregates are effectively diminished in MG-H1+HMWF treated groups

Answer:

We provided 4 extra H&E stain findings from 4 more mouse samples due to the reviewer's response deadline. We made corrections on P4, L149-154 as follows:

“The MG-H1 group may suggest the lymphoid aggregate in colon tissue which means infiltration of neutrophils (black arrow). The lymphoid aggregates might be effectively diminished in the MG-H1+HMWF treated groups. It should be noted, however, that H&E staining is not conclusive for neutrophils, and no staining for a neutrophil specific marker such as RB6-8C5 was conducted in this study. The current data is qualitative in nature.”

Finally, we really appreciate the reviews’ critical and valuable comments which have been guided in revising our manuscript. We believe that our manuscript has been much improved due to the revision based on the reviewer’s suggestions.

Reviewer 2 Report

I don't have a question

Author Response

First of all, again we’d like to express our sincere gratitude to the reviewer for the careful and critical reading of our manuscript.

Reviewer 3 Report

The authors have addressed the raised comments from previous revision as possible. The quality of the research has been also improved and can be accepted in the present form. However, there is still something wrong between the title and authors name and affiliation in the main and supplementary file. Please revise this problem, otherwise there is a conflict of interest between co-authors! 

Author Response

Reviewer(s)' Comments to Author:

Reviewer 3:

Comments and Suggestion for Authors

Comments:

The authors have addressed the raised comments from previous revision as possible. The quality of the research has been also improved and can be accepted in the present form. However, there is still something wrong between the title and authors name and affiliation in the main and supplementary file. Please revise this problem, otherwise there is a conflict of interest between co-authors!

> Answer:

First of all, again We thank the reviewer for the help and attentive comments on our manuscript.

With highlighted in green color text, we made the correction in response.

We corrected the title, authors' names, and affiliations in both the main file and the supplementary file as the reviewer had advised.

Please see on page (P) 1, lines (L) 1-10 in the main text file and in the supplementary file as follows:

In the main text file,

High molecular weight fucoidan restores intestinal integrity by regulating inflammation and tight junction loss induced by methylglyoxal-derived hydroimidazolone-1

Jae-Min Lim 1, Hee Joon Yoo 1 and Kwang-Won Lee 1,2*

  • Department of Biotechnology, College of Life Science and Biotechnology, Korea University, Seoul 02841, Republic of Korea
  • Department of Food Bioscience and Technology, College of Life Science and Biotechnology, Korea University, Seoul 02841, Republic of Korea

     *    Correspondence: kwangwon@korea.ac.kr; Tel.: +82-2-3290-3027

 In the supplementary file,

High molecular weight fucoidan restores intestinal integrity by regulating inflammation and tight junction loss induced by methylglyoxal-derived hydroimidazolone-1

Jae-Min Lim 1, Hee Joon Yoo 1 and Kwang-Won Lee 1,2*

  • Department of Biotechnology, College of Life Science and Biotechnology, Korea University, Seoul 02841, Republic of Korea
  • Department of Food Bioscience and Technology, College of Life Science and Biotechnology, Korea University, Seoul 02841, Republic of Korea

     *   Correspondence: kwangwon@korea.ac.kr; Tel.: +82-2-3290-3027

Finally, we really appreciate the review’s critical and valuable comments which have been guided in revising our manuscript. We believe that our manuscript has been much improved due to the revision based on the reviewer’s suggestions.
